# Lysyl Oxidase-like Protein Recognizes Viral Envelope Proteins and Bacterial Polysaccharides against Pathogen Infection via Induction of Expression of Antimicrobial Peptides

**DOI:** 10.3390/v14092072

**Published:** 2022-09-18

**Authors:** Peng-Yuan Lu, Guo-Juan Niu, Pan-Pan Hong, Jin-Xing Wang

**Affiliations:** 1Shandong Provincial Key Laboratory of Animal Cells and Developmental Biology, School of Life Sciences, Shandong University, Qingdao 266237, China; 2State Key Laboratory of Microbial Technology, Shandong University, Qingdao 266237, China

**Keywords:** *Marsupenaeus japonicus*, white spot syndrome virus, *Vibro anguillarum*, lysyl oxidase-like, Dorsal, pattern recognition receptor

## Abstract

Lysyl oxidases (LOXs) are copper-dependent monoamine oxidases, and they play critical roles in extracellular matrix (ECM) remodeling. The LOX and LOX-like (LOXL) proteins also have a variety of biological functions, such as development and growth regulation, tumor suppression, and cellular senescence. However, the functions of LOXLs containing repeated scavenger receptor cysteine-rich (SRCR) domains in immunity are rarely reported. In this study, we characterized the antiviral and antibacterial functions of a lysyl oxidase-like (LOXL) protein containing tandem SRCR domains in *Marsupenaeus japonicus*. The mRNA level of *LoxL* was significantly upregulated in the hemocytes and intestines of shrimp challenged using white spot syndrome virus (WSSV) or bacteria. After the knockdown of *LoxL* via RNA interference, WSSV replication and bacterial loads were apparently increased, and the survival rate of the shrimp decreased significantly, suggesting that LOXL functions against pathogen infection in shrimp. Mechanistically, LOXL interacted with the envelope proteins of WSSV or with lipopolysaccharide and peptidoglycan from bacteria in shrimp challenged using WSSV or bacteria, and it promoted the expression of a battery of antimicrobial peptides (AMPs) via the induction of Dorsal nuclear translocation against viral and bacterial infection. Moreover, LOXL expression was also positively regulated by Dorsal in the shrimp challenged by pathogens. These results indicate that, by acting as a pattern recognition receptor, LOXL plays vital roles in antiviral and antibacterial innate immunity by enhancing the expression of AMPs in shrimp.

## 1. Introduction

Lysyl oxidases (LOXs) and LOX-like proteins are copper-dependent monoamine oxidases that oxidize primary amine substrates to reactive aldehydes in the extracellular matrix (ECM) and play a central role in ECM remodeling [1]. Five members in the LOX family are identified in mammals, including LOX, LOX-like 1 (LOXL1), LOXL2, LOXL3, and LOXL4 [2,3,4]. These five members share a highly conserved carboxy terminus, which contains a lysyl oxidase domain, an unconserved amino terminus, which contains a signal peptide, and four repeats of scavenger receptor cysteine-rich (SRCR) domains in LOXL2-4 [5,6].

The structural similarity of the highly conserved lysyl oxidase domains among the LOX and LOX-like proteins results in similar amine oxidase activities, which are well known for their function of cross-linking collagen and elastin to remodel the ECM in animals [7,8,9,10,11]. A growing body of evidence indicates that the LOX family is involved in the regulation of the proliferation, migration, invasion, and metastasis of cancer cells [12]. The current results reveal that LOX family members participate in inflammatory pathways, which are important processes in the immune response [13,14]. In addition, LOX and LOXL proteins also play enzyme-activity-independent roles in extracellular, intracellular, and nuclear environments, the mechanistic details of which remain to be fully characterized.

LOXL proteins share a common specific structure, the SRCR domains at their N-terminus. SRCR domains are ancient and highly conserved domains characterized by three or four intradomain disulfide bonds that are present in soluble and membrane proteins associated with the pattern recognition of pathogen-associated molecular patterns (PAMPs) [15]. Homologs of LOX-like proteins (LOXL1 and LOXL2, with one or two SRCR domains) are identified in *Drosophila melanogaster* and *Anopheles gambiae* and are classified as class A scavenger receptors (SRA) [16], but the functions of the receptors are unclear. In humans, members of the class A scavenger receptor family are important pattern recognition receptors (PRRs) that play important functions in innate immunity and inflammatory responses [17]. The MARCO (macrophage receptor with collagenous structure) with the SRCR domain, a member of the class A scavenger receptor family, can be readily upregulated under the conditions of bacterial infection, and it has the ability to bind to both Gram-positive and Gram-negative bacteria [18,19]. Several scavenger receptors act as co-receptors of the Toll-like receptor (TLR) and modulate the immune response against invading pathogens [20].

The expression of LOX is regulated by several cytokines in inflammatory pathways. LOX upregulation is closely associated with tumor necrosis factor α (TNF-α), a well-known pro-inflammatory cytokine. In cardiac fibroblasts of mammals, TNF-α increases LOX expression through transforming growth factor-β (TGF-β) and PI3Kinase signaling pathways [21]. TNF-α also induces epithelial LOX expression by activating the nuclear factor NF-κB and TGF-β [22]. However, the function of TNF-α is pleiotropic; in diabetes, osteoporosis, and rheumatic arthritis, TNF-α inhibits the functions of LOX [14]. Several members of the interleukin (IL) family are also important molecules in the regulation of LOX expression in inflammatory pathways. The regulatory roles and mechanisms of IL members, which are recognized to affect LOXs, are currently less clear [14].

In this study, we identified a lysyl oxidase-like (LOXL) protein containing two SRCR domains in kuruma shrimp, *Marsupenaeus japonicus*. RNA knockdown analysis revealed that LOXL possessed antiviral and antibacterial functions in shrimp. Further studies indicated that LOXL acted as a PRR and promoted the expression of antimicrobial peptides (AMPs) via the induction of the nuclear translocation of Dorsal, the NF-κB transcription factor of the Toll pathway.

## 2. Materials and Methods

### 2.1. Shrimp Challenge and Sample Collection

Shrimp (8–12 g each) used in our experiment were purchased from an aquatic product market in Qingdao, Shandong province, China, and cultured in aerated seawater at 22 °C. Fifty microliters of WSSV (1 × 10^5^ virions) (isolated from WSSV-infected *Procambarus clarkii* following our previous method [23]) was injected into the penultimate segment of the tails of the shrimp. For the bacterial challenge, shrimp were injected with *Vibro anguillarum* (3 × 10^7^ CFU/shrimp) at the same position. Different organs, including hearts, hepatopancreas, gills, stomachs, and intestines, were collected from at least three shrimp challenged with WSSV or *V. anguillarum*, for RNA and protein extraction. For hemocyte collection, hemolymph was extracted with a syringe, pre-filled with 0.8 mL anticoagulant buffer, from the abdominal sinus, and then quickly centrifuged at 500× *g* for 8 min at 4 °C to obtain the hemocytes.

### 2.2. RNA and Protein Extraction, and cDNA Synthesis

TRIzol (ET101, Transgen, Beijing, China) was used for the extraction of total RNA from the hemocytes and different organs of shrimp. Proteins from the different organs were extracted using a radio-immunoprecipitation assay (RIPA) buffer (50 mM Tris-HCl, 150 mM NaCl, 0.1% SDS, 0.5% Nonidet P-40, 1 mM EDTA, and 0.5 mM PMSF, pH 8.0). The first-strand cDNA was synthesized using a cDNA synthesis kit (ABM, Vancouver, BC, Canada) according to the manufacturer’s instructions.

### 2.3. Recombinant Expression and Purification

The DNA fragment encoding the LOXL protein was amplified using the primers (LOXL-ORF-F/R) listed in Appendix A. The restriction enzymes *Eco*RI and *Bam*HI were used to digest the purified PCR product, and then the product was ligated into the pGEX-4T-2 (+) vector. The recombinant plasmid was transformed into *Escherichia coli* JM109 cells. Protein expression was induced with 1 mM isopropyl-β-D-thiogalactopyranoside (IPTG) at 37 °C for 4 h. The resulting inclusion body was washed with buffer A (2M Tris-HCL and 0.1M EDTA) and centrifuged at 12,000× *g* for 10 min at 4 °C, and then washed with buffer B (2M Tris-HCL, 0.1M EDTA, and 2M urea) and centrifuged according to the above method. After repeating this three times, the precipitate obtained was dissolved in the denatured solution (2M Tris-HCL and 8M urea) and shaken at 37 °C for 1 h. The supernatant was placed into the dialysate and dialyzed for 48 h at 4 °C to acquire the refold recombinant LOXL protein, and then the refolded proteins were subjected to affinity chromatography using glutathione (GST) resin (C600031, BBI, Shanghai, China) to obtain purified proteins, following the manufacturer’s instructions. The refolded purified proteins were used for subsequent experiments. The expression of the envelope proteins of WSSV, the GST tag protein and the Trx-His tag protein were performed using previously constructed vectors in the laboratory.

### 2.4. Quantitative Real-Time RT-PCR

We performed RNA extraction and reverse transcription using the same methods as described above. The temporal expression of *LoxL* was detected via quantitative real-time PCR (qPCR) using the primers LOXL-RT-F and LOXL-RT-R (Appendix A) in a real-time thermal cycler. *β-actin* and *Ef1-α* amplification using Actin-RT-F/R and Ef1-α-RT-F/R (Appendix A) were used as the internal controls. QPCR was performed as follows: 1 cycle at 95 °C for 10 min, 40 cycles at 95 °C for 15 s and 60 °C for 1 min, and then at 78 °C for 2 s (for fluorescence signal acquisition). We calculated the obtained data using the cycle threshold (2^−ΔΔCT^) method. The expression levels of *Vp28*, *Dorsal*, *Relish*, and *STAT* were also detected using qPCR with primers of VP28-RT-F/R, Dorsal-RT-F/R, and Relish-RT-F/R (Appendix A). We also analyzed the expression levels of several AMPs (*Alf-a1*, *Alf-b1*, *Alf-c1*, *Alf-c2*, *Alf-e1*, *CrusI1*, and *CrusI3*) using qPCR with the primers listed in Appendix A.

### 2.5. Quantification of WSSV Copies

We quantified WSSV copy numbers using previously reported methods [23,24]. Briefly, a VP28 fragment of WSSV was amplified and inserted into the plasmid (pBlue-Script II SK+). The recombinant plasmid was quantified using a spectrophotometer (Gene-Quant, Amersham Biosciences, Piscataway, NJ, USA). The known molecular weight of the plasmid was used to calculate its copy number. Then, we gradually diluted the plasmid (10^9^, 10^8^, 10^7^, 10^6^, 10^5^, and 10^4^) and used the diluted plasmids of different concentrations as templates for qPCR with the primers VP28-RT-F and VP28-RT-R (Appendix A). The standard curve for WSSV quantification was made from the cycle threshold (CT) and the quantity of the template. The genomic DNA extracted from the WSSV infected tissues were analyzed via qPCR to obtain the absolute copies of WSSV from the infected tissues based on the data from gradient-diluted plasmid samples. 

### 2.6. RNA Interference

To analyze the function of LOXL, RNA interference (RNAi) was performed by injecting double-stranded RNA (dsRNA) into shrimp. We used the primers LOXL-Ri-F/R, Dorsal-Ri-F/R, Relish-Ri-F/R, STAT-Ri-F/R, and Gfp-Ri-F/R (Appendix A) with the sequence of the T7 promoter to amplify the template for the production of *dsLoxL*, *dsDorsal*, *dsRelish*, *dsSTAT*, and *dsGfp* with T7 RNA polymerase. Then, *dsLoxL*, *dsDorsal*, *dsRelish*, *dsSTAT*, or *dsGfp* (1 μg/μL) was injected into the penultimate segment of the shrimp tail (50 μL each), and another 50 μL dsRNA was injected into the shrimp 24 h later. After 24 h, RNA interference efficiency in the hemocytes and intestines was detected using qPCR. *β-actin* and *Ef1-α* were used as the internal references.

### 2.7. Survival Rate Assays

Survival rate assays were performed to further prove the function of LOXL. We divided the shrimp into two groups (n ≥ 25 shrimp per group). After injecting *dsLoxL* for 48 h, the shrimp were injected with WSSV (1 × 10^5^ copies) or *V. anguillarum* (3 × 10^7^ CFU/shrimp). The dead shrimp were monitored every 24 h after WSSV or bacterial injection, and *dsGfp* was used as the control. The survival rate of each group was calculated, and survival curves were statistically analyzed using GraphPad Prism 8.0 software.

### 2.8. Western Blot Analysis

We homogenized the collected tissues in RIPA buffer and centrifuged at 12,000× *g* for 10 min at 4 °C for supernatant collection. Proteins separated using 10% SDS-PAGE were transferred onto a nitrocellulose membrane using transfer buffer (25 mM Tris–HCl, 20 mM glycine, 0.037% mM SDS, and 20% ethanol). We blocked the membrane with 3% nonfat milk diluted in TBST (100 mM NaCl, pH 7.5, 10 mM Tris-HCl, and 0.02% Tween) for 30 min, shaking slowly at room temperature, and then the membrane was incubated overnight at 4 °C with primary antibodies: anti-Dorsal with 1:500 (prepared in our laboratory), anti-beta actin (ACTB) with 1:250 (prepared in our laboratory), and histone-3 (H3) polyclonal antibodies with 1:2500 (A2348, ABclonal, Wuhan, China). After washing with TBST three times, we incubated the membrane with HRP-conjugated secondary antibody (ZB2301 ZSGB-Bio, Beijing, China, 1:5000) for 3 h in 3% nonfat milk at room temperature with gentle shaking. We washed the membrane with TBST three times and then with TBS (10 mM Tris-HCl and 150 mM NaCl, pH 8.0) three times. We developed the target bands using enhanced chemiluminescence (ECL). β-Actin or histone-3 was used as the loading control.

### 2.9. Isolation of Nuclear and Cytoplasmic Proteins

Intestinal proteins were extracted from shrimp using a nuclear and cytoplasmic protein extraction kit (R0050, Solarbio, Beijing, China) following the manufacturer’s instructions. Intestinal protein was extracted from at least three shrimp in each experiment to reduce the individual differences.

### 2.10. Fluorescent Immunocytochemical Assay

The hemocytes were collected using the above-mentioned method. After washing three times with PBS, the collected hemocytes were dropped onto poly-l-lysine-coated slides and allowed to settle for 2 h at room temperature. The hemocytes on the slide were treated with 0.2% Triton X-100 and washed three times with PBS. After blocking with 3% BSA (PBS dilution) at 37 °C for 1 h, we incubated the hemocytes with 3% BSA-diluted antibody overnight at 4 °C. The second antibody, goat anti-rabbit Alexa 488 in 3% BSA, was added onto the slide and incubated for 2 h after washing three times with PBS. Then, the hemocytes were stained with DAPI for 10 min after washing three times with PBS. Finally, we washed the slides with PBS three times, and the hemocytes were observed under a fluorescence microscope (Olympus BX51, Olympus Corporation, Tokyo, Japan). The colocalization of Dorsal and DAPI-stained nuclei in the hemocytes was analyzed using CIF ImageJ software (NIH, Bethesda, MD, USA).

### 2.11. Pull-Down Assay

The interaction between the LOXL and WSSV envelope proteins was explored using pull-down assays. We recombinantly expressed the four main envelope proteins of WSSV (VP19, VP24, VP26, and VP28) in *E. coli* with the recombinant vector pET32A-VPs. Purified GST-tagged LOXL was incubated with each of the four His-tagged envelope proteins (1:1) overnight at 4 °C on a flipping shaker. After incubation with GST-bound resin (150 μL) for 45 min at 4 ˚C on a flipping shaker, we washed the resin with PBS five times. The bound proteins were eluted using elution buffer (10 mM reduced glutathione and 50 mM Tris-HCl, pH 8.0). The eluted proteins were analyzed using 10% SDS-PAGE. Purified His-tag protein and GST-tag protein were used as the controls.

### 2.12. Bacterial Clearance Assay

To analyze the function of LOXL in shrimp, a bacterial clearance assay was carried out. Shrimp were injected with *V. anguillarum* (3 × 10^7^ CFU/shrimp) at 24 h post knockdown of *LoxL*, and the hemocytes and intestines of the shrimp were collected at 3 h post injection (hpi). The genomic DNA were extracted from the bacteria-infected tissues, and 16S rRNA gene copies were determined using universal 16S rRNA primers (Appendix A) via qPCR. The hemolymph and intestines were also collected at 3 hpi and diluted 10,000-fold with sterile PBS. The diluted hemolymph and intestines were put on Luria–Bertani (LB) (1% tryptone, 0.5% yeast extract, and 1% NaCl) agar plates and cultured overnight at 37 °C, and the bacterial colonies on LB agar plates were counted. This assay was repeated three times.

### 2.13. Polysaccharide Binding Assay

An enzyme-linked immunosorbent assay (ELISA) reported previously [25] was used to detect the direct binding activity of LOXL to polysaccharides, including lipopolysaccharides (LPS) and peptidoglycan (PGN) (Sigma, St. Louis, MO, USA). Eighty micrograms of polysaccharides were dissolved in one milliliter of water and sonicated for 45 s. Then, 50 μL of the polysaccharides was added to each well of the 96-well polypropylene microtiter plate, incubated at 37 °C overnight, and heated at 60 °C for 30 min. The wells of the plate were blocked by BSA (1 mg/mL, 200 μL) at 37 °C for 2 h. After washing four times with TBS (200 μL), the purified LOXL proteins (in TBS containing 0.1 mg/mL BSA) at a concentration of 2 mg/mL were added to the plate wells and incubated for 3 h at room temperature. One hundred microliters of diluted 1:1000 mouse anti-GST antibody (in 0.1 mg/mL BSA) was added to each well of the plate and incubated at 37 °C for 1 h. After washing with TBS, alkaline phosphatase-conjugated goat anti-mouse IgG diluted 1:3000 (in 0.1 mg/mL BSA) was added to each well and incubated for 1 h at 37 °C. The plates were washed four times with TBS, and then the colors were developed by adding 100 μL p-nitro-phenyl phosphate (in 10 mM diethanolamine and 0.5 mM MgCl_2_) at a concentration of 1 mg/mL. The purified recombinant GST protein was used as a control. The absorbance was read at 450 nm. The assay was repeated three times. GraphPad Prism version 8.0 software was used to calculate the dissociation constants (K_d_) and maximum binding (B_max_) parameters.

### 2.14. Bacterial Binding Assay

The binding activity of LOXL to bacteria was detected using Gram-positive bacteria (including *Bacillus thuringiensis*, *Bacillus subtilis*, *Bacillus megaterium*, and *Staphylococcus aureus*) and Gram-negative bacteria (including *Pseudomonas aeruginosa*, *E. coli*, *Klebsiella peneunoniae*, and *V. anguillarum*). The bacteria cultured in LB medium were collected by centrifugation at 5000× *g* for 5 min and washed twice with TBS. Then the bacteria were adjusted to 1.0 OD 600 with TBS. Ten microliters of LOXL protein at a concentration of 2 mg/mL was added to 200 μL TBS containing bacteria and incubated for 30 min at room temperature. The bacteria were collected by centrifugation, washed three times with TBS, and then eluted with 7% SDS. The eluted solution and deposited bacteria were used for Western blot analysis to test the bacterial binding ability, using anti-GST as the first antibody.

### 2.15. Statistical Analysis

Experimental data used for statistics were the mean ± standard deviation (SD) of at least 3 replicates. Significant difference was analyzed by Student’s *t*-test for paired comparisons, or by one-way ANOVA for multiple comparisons. The *p* values in paired comparisons are marked in the figures. In the ANOVA analysis, different lowercase letters indicate significant differences (*p* < 0.05). The survival rate of each group was calculated, and the survival curves are presented as Kaplan–Meier plots and also statistically using a log-rank test. GraphPad 8.0 data view software was used to produce all the statistical analyses, and ImageJ software (National Institutes of Health, http//imagej.nih.gov/ij/download.html (accessed on 28 September 2021)) was used to analyze the Western blotting bands from three independent experiments.

## 3. Results

### 3.1. LOXL Is Upregulated Significantly in Shrimp Challenged with WSSV or V. anguillarum

The LOX-like protein (LOXL) identified in *M. japonicus* contained a signal peptide, two scavenger receptor cysteine-rich (SRCR) domains at the amino terminal, and a lysyl oxidase domain at the carboxyl terminal. The domain architectures of shrimp LOXL are similar to *Homo sapiens* LOXL2-4, which contain SRCR domains but differ in the domain numbers (Appendix A). The results of phylogenetic analysis showed that LOXL of *M. japonicus* was clustered into a clade with LOXL2 in *Penaeus vannamei* (Appendix A). Sequence alignment analysis revealed that, although the SRCR domains shared different levels of homology, the relative positions of the six cysteines are well-conserved within the domain of LOXLs in *D. melanogaster*, *P. vannamei*, *P. clarkii*, and *M. japonicus* (Appendix A).

Tissue distribution analysis indicated that *LoxL* was constitutively expressed in the hemocytes, hearts, hepatopancreas, gills, stomachs, and intestines at the mRNA level (Figure 1A). The temporal and spatial expression patterns of *LoxL* after WSSV challenge were analyzed using qPCR. The results showed that *LoxL* was significantly upregulated from 24 to 48 hpi in hemocytes, upregulated at 3 hpi, recovered to basal level from 6 to 24 hpi, and then upregulated at 48 hpi in intestines (Figure 1B,C). We also analyzed the temporal and spatial expression patterns of *LoxL* in shrimp after *V. anguillarum* challenge via qPCR. The results showed that the mRNA level of *LoxL* was significantly upregulated in hemocytes and intestines (Figure 1D,E). These results suggest that LOXL might participate in the immune response to WSSV and *V. anguillarum* infection in shrimp.

### 3.2. LOXL Inhibits WSSV Replication and Bacterial Proliferation in Shrimp

To explore the function of LOXL in vivo, RNA interference was performed, and the replication of WSSV was detected in hemocytes and intestines via qPCR and Western blotting, using VP28 (WSSV envelope protein) as an indicator. The results showed that the mRNA level of *LoxL* was reduced significantly after the injection of *dsLoxL* at 48 hpi (Figure 2A). After the injection of WSSV into the shrimp, the expression of VP28 in the *LoxL*-RNAi group was significantly higher than that in the *Gfp*-RNAi group at 24 hpi of WSSV, as analyzed using qPCR and Western blot (Figure 2B,C1,C2). Meanwhile, the immediate early gene of the WSSV (*Ie1*) and WSSV copies in the intestines of the shrimp were increased markedly (Figure 2D,E). To further confirm the results, a survival rate assay was carried out and the results showed that the survival rate of *LoxL*-RNAi shrimp infected with WSSV decreased significantly compared with the *Gfp*-RNAi group (Figure 2F). The load of bacteria in the *LoxL*-RNAi shrimp infected by *V. anguillarum* was also analyzed. After knocking down *LoxL*, we infected the shrimp with *V. anguillarum*, and the bacteria load were detected at 3 hpi; the results showed that the bacterial load was significantly increased in the hemolymph and intestines (Figure 2G–I), and the survival rate of the RNAi-shrimp declined significantly (Figure 2J). All of the results suggest that LOXL possesses antiviral and antibacterial functions in shrimp.

### 3.3. LOXL Promotes the Expression of Antimicrobial Peptides

Three main immune-related signal pathways—Toll and immune deficiency (IMD) and JAK/STAT signaling—are involved in antiviral and antibacterial responses via the regulation of the transcription of antimicrobial peptides (AMPs) in shrimp [26]. A previous study found that the expression of AMPs, including *Alf-b1*, *Alf-c2*, *CrusI1*, and *CrusI3*, was regulated by Toll/Dorsal signaling [27]; *Alf-c1* and *Alf-e1* are regulated by IMD/Relish signaling [28]; and the JAK/STAT pathway is related to the expression of *Alf-a1*, *Alf-c1*, *Alf-c2*, and *CruΙ-1* [29]. To study the possible mechanisms of LOXL against pathogens in shrimp, we first detected the expression of transcription factors (Dorsal, Relish, and Stat) and AMPs in *LoxL*-RNAi shrimp. The results showed that, after the knockdown of *LoxL* (Figure 3A), the expression of *Dorsal*, *Alf-b1*, *Alf-c2*, *CrusI1*, and *CrusI3* was significantly decreased in the *LoxL*-RNAi group compared with the *dsGfp* injection group (Figure 3B–D). The expression of *Relish* was slightly increased in hemocytes (Figure 3B) and intestines (Figure 3C), and the expression of *STAT* was slightly decreased in hemocytes (Figure 3E) and intestines (Figure 3F), but there was no significant difference. Meanwhile, the expression of *Alf-a1, Alf-c1*, and *Alf-e1* was not affected by the knockdown of *LoxL* in hemocytes (Figure 3D) and intestines (Figure 3E). All of the results suggest that LOXL might be involved in the activation of Toll/Dorsal signaling to promote AMP expression.

### 3.4. LOXL Promotes the Nuclear Translocation of Dorsal

In order to confirm that LOXL inhibits WSSV replication or bacterial proliferation via Toll/Dorsal signaling, the subcellular distribution of the NF-κB-like transcription factor, Dorsal, was detected using Western blot after knocking down *LoxL* (Figure 4A) in shrimp infected with WSSV. The results indicated that Dorsal content in the nucleus of the intestine cells of the *LoxL* knockdown group was distinctly decreased compared with that of the *Gfp*-RNAi group at 2 h post WSSV infection (Figure 4B1,B2). To further confirm that LOXL is involved in the activation of Dorsal signaling, a fluorescent immunocytochemical assay was performed, and the results showed that the nuclear translocation of Dorsal was inhibited in the hemocytes of the shrimp at 2 h post WSSV infection (Figure 4C1,C2). All of the results suggest that LOXL promotes the nuclear translocation of Dorsal in shrimp challenged with WSSV.

### 3.5. LOXL Interacts with Envelope Proteins of WSSV

As a member of the scavenger receptor A, LOXL might play its function as a PRR. To investigate the interaction of LOXL with WSSV, pull-down assays were performed to explore the interactions of LOXL with four envelope proteins of WSSV, including VP19, VP24, VP26, and VP28 in vitro. The recombinant proteins were expressed in *E. coli* (Figure 5A,B). The results of the GST pull-down showed that the LOXL interacted with four envelope proteins (Figure 5C–F). Trx-His protein purified from *E. coli* (Figure 5B) did not interact with LOXL (Figure 5G). We also performed pull-down assays using GST, with VP protein as a control, which showed that the GST and VP proteins did not interact with each other (Figure 5H–K). The results suggested that LOXL could recognize WSSV via interactions with its envelope proteins.

### 3.6. LOXL Recognizes WSSV through the SRCR Domains

To further investigate which domain of LOXL was responsible for the interaction with envelope proteins, two SRCR domains (SRCR1 and SRCR2) in LOXL were expressed and purified, respectively, for the pull-down assays (Figure 6A). The results of the GST pull-down showed that SRCR1 interacted with VP19 (Figure 6B), VP24 (Figure 6C), and VP28 (Figure 6D), but not with VP26 (Figure 6E). Because of the similar molecular masses of SRCR and VP24, the interaction of the two proteins was also analyzed via Western blot using the GST antibody (Figure 6F) and the His-tag antibody (Figure 6G), and the same results were obtained. However, SRCR2 interacted with all four kinds of envelope proteins of WSSV (Figure 6H–K). We also used anti-GST and anti-His antibodies to detect the interaction between SRCR2 and VP24 (Figure 6L,M). All of the results suggested that the SRCR domains of LOXL were responsible for the interaction with WSSV. In addition, we generated the purified lysyl oxidase domain of LOXL and performed pull-down assays to verify the interaction with VP proteins, and the results showed that the lysyl oxidase domain could not interact with the VP proteins of WSSV (Appendix A). All of the results suggest that LOXL works as a PRR via its SRCR domains.

### 3.7. LOXL Binds to Bacteria via Interactions with LPS and PGN

To analyze the interaction of LOXL with bacteria, bacterial binding assays were performed using different bacteria, including Gram-positive bacteria (*B. thuringiensis*, *B. subtilis*, *B. megaterium*, and *S. aureus*) and Gram-negative bacteria (*P. aeruginosa*, *E. coli*, *K. peneunoniae*, and *V. anguillarum*). As shown in Figure 7A, the interactions of rLOXL with different bacteria were found using the elution solution via SDS (Figure 7A, 7% SDS), and no strong binding activity to the bacteria was detected after using the deposited bacteria after 7% SDS elution (Figure 7A, bacteria). To further analyze the PAMPs of bacteria that rLOXL bound with, we performed ELISA to detect direct binding activity for the polysaccharides, including LPS and PGN. The results showed that LOXL bound to LPS and PGN (Figure 7B,C). All of the results suggest that LOXL recognized bacteria by binding to LPS and PGN on the bacterial surface.

### 3.8. LOXL Expression Is Regulated by Dorsal Signaling

Previous studies on mammals have demonstrated that the expression of LOX is regulated by different signal pathways. To analyze the relationship of LOXL with Toll/Dorsal signaling, we knocked down Dorsal and detected the expression of *LoxL* in hemocytes and intestines by qPCR. The results showed that the mRNA level of *LoxL* was reduced significantly after the injection of *dsDorsal* at 48 hpi (Figure 8A,B). However, after the knockdown of Relish via the injection of *dsRelish* (Figure 8C), no significant change was detected for *LoxL* mRNA levels (Figure 8D). The same results were obtained after the knockdown of STAT (Figure 8E,F). We further obtained the promoter sequence of the shrimp *LoxL* using Nucleotide BLAST from the *M*. *japonicus* genome. The bioinformatic analysis using the *LoxL* promoter sequence revealed putative binding sites for transcription factor NF-κB in the regions between −612 and −601 bp, and between −489 and −479 bp of the *LoxL* promoter. Interestingly, we also identified a putative binding site of STAT in the region between −360 and −350 bp (Appendix A). The results suggested that there might be a positive feedback loop for *LoxL* expression in shrimp. The interaction of Dorsal with the promoter needs further study.

## 4. Discussion

In the present study, we discovered that a LOXL was involved in the innate immunity of shrimp working as a PRR. After WSSV or *V. anguillarum* challenge, LOXL was significantly upregulated in the hemocytes and intestines of the shrimp. LOXL interacted with the envelope proteins of WSSV or with the polysaccharides of bacteria and promoted the expression of AMPs via the induction of nuclear translocation of Dorsal against bacterial and viral infection. 

As the first and best characterized PRR family, Tolls and Toll-like receptors (TLRs) play indispensable roles in innate immunity by recognizing and responding to multiple microbial pathogens [30,31,32]. Some scavenger receptors act as the co-receptors of TLRs in the recognition of PAMPs and in the activation of downstream signaling transducers. For example, the class A scavenger receptors, including SR-A1, MARCO, and a class B scavenger receptor, CD36, have been determined as co-receptors of TLRs [33,34]. In invertebrates, a class B receptor OoSR-B from *Octopus ocellatus* was also reported as a co-receptor of TLR, participating in the recognition of bacteria [35]. The Toll signaling pathway has been identified in shrimp, and many components of the pathway have been well-studied, such as Tolls, Spätzle, MyD88, TRAF6, Dorsal, Cactus, etc. [27,36]. In this signaling pathway, the binding of ligands to TLRs causes a series of signaling cascades, which ultimately leads to the activation of Dorsal, inducing the expression of related AMPs such as *Alf-b1*, *Alf-c2*, *CrusI1*, and *CrusI3* [26,37]. Three families of AMPs—including penaeidins, crustins, and anti-lipopolysaccharide factors, possessing antibacterial, antifungal, and antiviral activities against different strains of bacteria, fungi, and viruses—are identified in shrimp; the expression of the AMPs are mainly regulated by the Toll and immune deficiency (IMD) and Janus kinase/signal transducers and activators of transcription (JAK/STAT) signal pathways [38,39]. In our study, we discovered that LOXL regulated the expression of AMPs, including *Alf-b1*, *Alf-c2*, *CrusI1*, and *CrusI3*, by inducing the nuclear translocation of Dorsal to defend against bacterial and WSSV infection, but not by the IMD or JAK/STAT pathways. Therefore, LOXL might work as a co-receptor to initiate the Toll signaling pathway during antibacterial and antiviral responses in shrimp. The interaction of LOXL with TLRs needs further investigation.

LOXs containing SRCR domains are classified as class A scavenger receptor family members in *D. melanogaster* and *A*. *gambiae* [16]. Five SRCR-containing proteins and four orthologous pairs exist both in the fruit fly and in the mosquito, but the function of LOXs in the insects is not clear. All proteins in the SRCR superfamily (SRCR-SF) contain the SRCR domains (the common feature in this group of proteins) [40]; therefore, LOXLs are also members of the SRCR-SF. As a conserved protein module, SRCR domains possess approximately 90 to 110 amino acids and are characterized by their highly conserved cysteine residues. According to the characteristics of their SRCR domains, two types of SRCR-SF members are reported: type A domains, which contain six cysteine residues, and type B domains, containing eight cysteine residues [15]. The SRCR domains in shrimp LOXL belong to the type A domains. Several studies suggest that the SRCR-SF members have been involved in the development of the immune system and in the regulation of immune responses [40]. In our study, we found that LOXL interacted with the envelope proteins of WSSV via the SRCR domains. In addition, the results of polysaccharide binding assays and bacterial binding assays showed that LOXL bound to Gram-positive and Gram-negative bacteria by interacting with LPS as well as PGN, indicating that LOXL functions as a PRR through the SRCR domains to recognize pathogens and thereby activate immune-related signaling pathways. 

The most fundamental function of the LOX family is to promote collagen cross-linking, which determines the ECM stability and promotes the deposition of collagen fibers in the ECM [13]. Thus, these cross-linking enzymes are implicated in many aspects of physiopathology, such as tumor progression, fibrosis, and cardiovascular disease, leading to vascular, cardiac, pulmonary, dermal, placenta, diaphragm, kidney, and pelvic floor disorders [14]. LOX has been found to be involved in inflammatory pathways; higher concentrations of TNF-α increased LOX expression via PI3K/Akt and Smad3 signaling [21] and TGF-β-mediated signaling [22]. In mammals, NF-κB is demonstrated as an indispensable transcription factor to mediate LOX repression by binding to a NF-κB binding site in the LOX promoter [41]. It is interesting to note that the expression of LOXLs regulated by cytokines via signaling is made in a cell-context-dependent manner. For example, IL-1β inhibits the expression of LOX in amnion fibroblasts [41] but stimulates the expression of LOX expression in collateral ligament fibroblasts [42]. In the present study, we found that the expression of LOXL decreased after RNA interference of Dorsal, indicating that LOXL expression was positively regulated via Dorsal signaling. Therefore, there might be a positive feedback loop for LOXL expression in shrimp challenged by pathogens.

The LOXs and LOXLs are reported as involved in fibrosis, tumorigenesis, and metastasis, and have been identified as therapeutic targets in human diseases [12]. Here, we found that, as a member of SRCR-SF [15], LOXL was also involved in pathogen recognition and modulation of immune response; this might indicate the possibility that targeting the protein could lead to diagnostic and therapeutic benefits for some pathologic states in humans. 

In conclusion, LOXL functioned as a PRR via binding PAMPs and activating Toll/Dorsal signal pathways to regulate the expression of AMPs against bacterial and viral infection in shrimp (Figure 9).

## Figures and Tables

**Figure 1 viruses-14-02072-f001:**
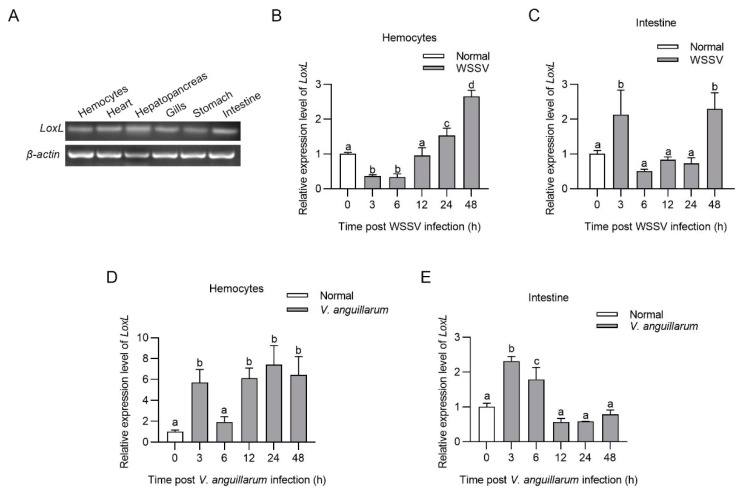
LOXL was upregulated in shrimp challenged with WSSV or *V. anguillarum*. (**A**) Tissue distribution of *LoxL* at mRNA level, detected using RT-PCR. *β-actin* was used as internal reference. (**B**,**C**) The mRNA expression patterns of *LoxL* in hemocytes (**B**) and intestines (**C**) after WSSV injection, detected using qPCR. Expression of *β-actin* and *Ef1-α* served as internal reference genes and were processed using geometric means. (**D**,**E**) The mRNA expression patterns of *LoxL* in hemocytes (**D**) and intestines (**E**) of shrimp after *V. anguillarum* injection, detected using qPCR. Expression of *β-actin* and *Ef1-α* served as internal reference genes and were processed using geometric means. The value of relative mRNA expression levels of *LoxL*/(*β-actin* × *Ef1-α*) were expressed as mean ± SD, and the value of the control shrimp was set as 1. Error bars in the figure represent SDs (three replicates). One-way ANOVA was used for statistical difference analysis. The significant differences are indicated using different lowercase letters (*p* < 0.05) in the ANOVA analysis.

**Figure 2 viruses-14-02072-f002:**
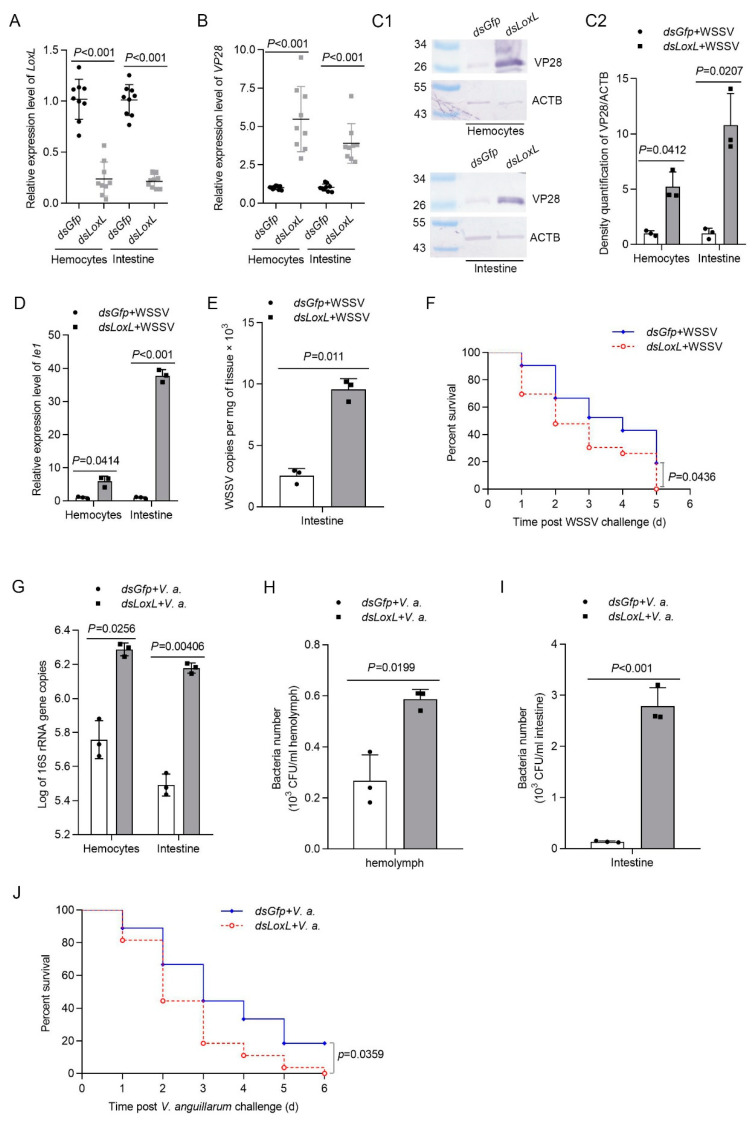
LOXL participates in the inhibition of WSSV replication and bacterial proliferation in shrimp. (**A**) Efficiency of *LoxL*-RNAi in the hemocytes and intestines detected via qPCR; *dsGfp* injection was used as the control. (**B**,**C1**) Detection of mRNA and protein expression levels of VP28 using qPCR (**B**) and Western blot (**C1**) at 24 hpi after WSSV injection. The shrimp were injected with WSSV after *dsLoxL* injection. Equal amounts of *dsGfp* injection shrimp were used as control. (**C2**) Statistical analysis of panel (**C1**) after the bands were digitalized using ImageJ software, based on three independent replicate experiments. (**D**) The mRNA expression of *Ie1* in *dsLoxL*-knockdown shrimp, detected using qPCR at 2 h after WSSV injection. (**E**) Detection of WSSV copies in the intestines via qPCR at 24 hpi after WSSV injection; *dsGfp* was used as the control, and the geometric value of *β-actin* and *Ef1-α* served as internal control. The values of mRNA relative expression levels were expressed as mean ± SD, and that of the control shrimp was set as 1. Error bars in the figure represent SDs (three replicates). (**F**) The survival rate of *LoxL*-RNAi shrimp challenged with WSSV; *dsGfp* was used as the control. (**G**) Detection of 16S rRNA gene copies in hemocytes and intestines via qPCR at 3 hpi after *V. anguillarum* injection; *dsGfp* injection was used as the control, and the geometric values of *β-actin* and *Ef1-α* served as internal control. The mRNA relative expression levels were expressed as the mean ± SD, and the value of the control shrimp was set as 1. Error bars in the represent indicate SDs (three replicates). (**H**,**I**) The hemolymph or content of intestines was collected at 3 hpi of *V. anguillarum* injection and cultured on LB agar plates. The number of bacterial colonies of hemolymph and intestines was counted on the plates after overnight culture. (**J**) The survival rate of *LoxL*-RNAi shrimp challenged with *V. anguillarum*. *dsGfp* injection was used as the control. Significant differences were analyzed using Student’s t-test; *p* value is shown in figures, and statistical significance was accepted at *p* < 0.05. The survival rate curves of each group are presented as Kaplan–Meier plots. The log-rank test in GraphPad Prism 8.0 software was used to analyze differences between the two groups.

**Figure 3 viruses-14-02072-f003:**
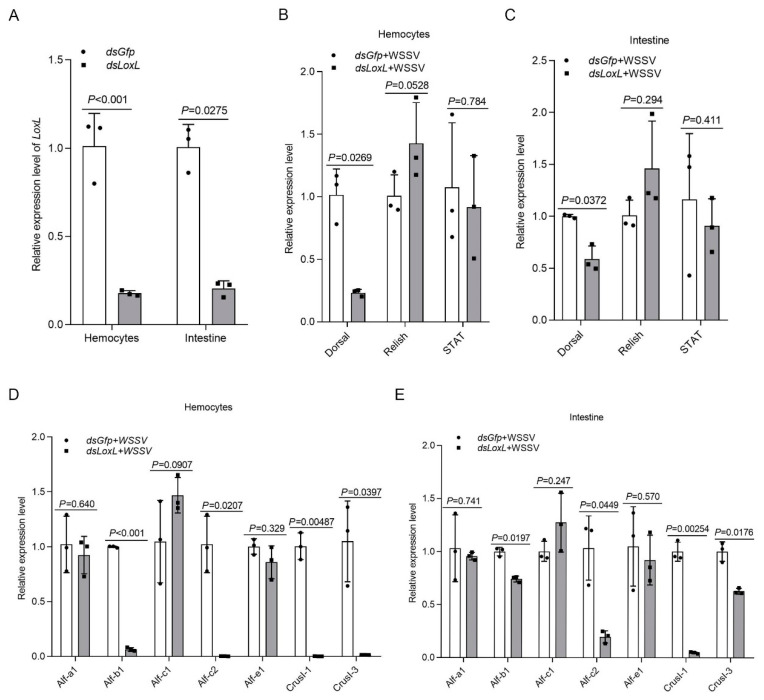
LOXL promotes the expression of *Amps* via activation of Dorsal signaling pathway. (**A**) Efficiency of *LoxL*-RNAi in hemocytes and intestines, detected using qPCR; *dsGfp* injection was used as the control. (**B**,**C**) Detection of mRNA levels of *Dorsal*, *Relish*, and *STAT* in hemocytes (**B**) and intestines (**C**) of *LoxL*-RNAi shrimp via qPCR at 24 hpi of WSSV injection; *dsGfp* injection was used as the control. (**D**,**E**) The expression of different *Amps* hemocytes (**D**) and intestines (**E**) of *LoxL*-knockdown shrimp; *dsGfp* injection was used as the control. Expression of *β-actin* and *Ef1-α* served as internal reference genes and were processed using geometric means. The values of mRNA relative expression levels were expressed as mean ± SD, and that of the control shrimp was set as 1. Error bars in the figure represent SDs of three replicates.

**Figure 4 viruses-14-02072-f004:**
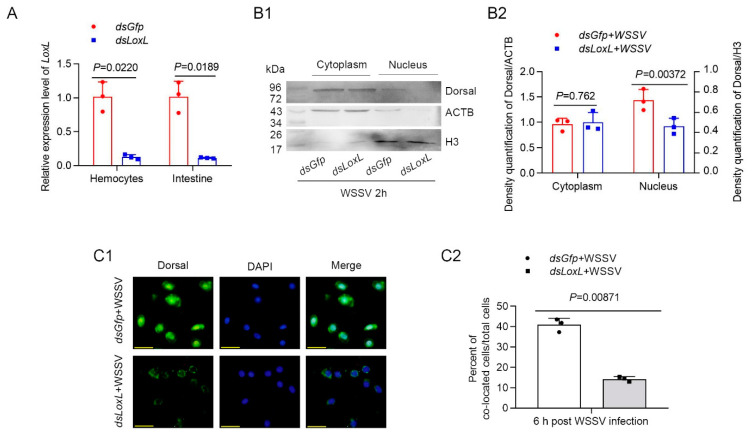
LOXL promotes expression and nuclear translocation of Dorsal in hemocytes and intestines of shrimp. (**A**) Efficiency of *LoxL*-RNAi in hemocytes and intestines of shrimp detected via qPCR; *dsGfp* injection was used as the control. (**B1**) The nuclear translocation of Dorsal was detected using Western blotting in the intestines of shrimp at 2 hpi; ACTB and H3 were used as loading controls. (**B2**) Statistical analysis of panel B after the bands were digitalized using ImageJ software on the basis of three independent experiments. (**C1**) The nuclear translocation of Dorsal in hemocytes of *LoxL*-RNAi shrimp at 6 hpi of WSSV challenge by fluorescent immunocytochemical assays; *dsGfp* injection was used as the control. Scale bar = 20 μm. (**C2**) Statistical analysis of nucleus–Dorsal co-localized cells compared with hemocytes. Significant differences were analyzed using a Student’s *t*-test; *p* value is shown in figures, and statistically significant differences were accepted at *p* < 0.05.

**Figure 5 viruses-14-02072-f005:**
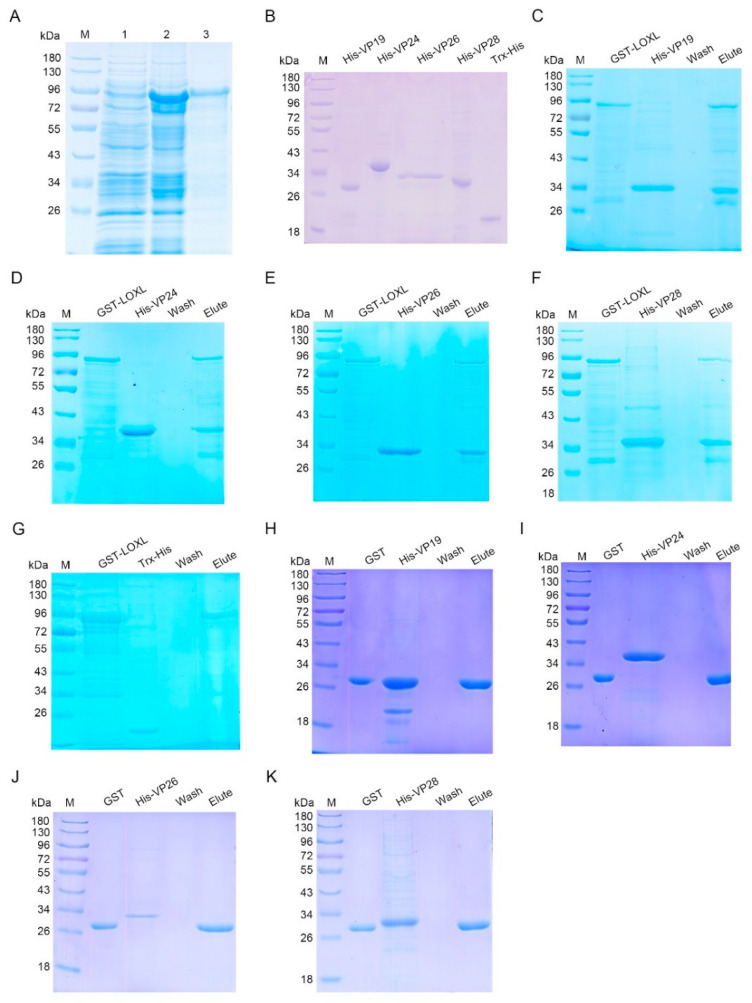
LOXL interacted with envelope proteins of WSSV. (**A**) Recombinant expression of LOXL in *E. coli*. Lane 1, the total proteins of *E. coli* with *LoxL*-pGEX4T-2. Lane 2, the total proteins of *E. coli* after the induction of IPTG. Lane 3, purified recombinant LOX protein. M, protein molecular mass markers. (**B**) Purified recombinant VP19, rVP24, rVP26, rVP28, and Trx-His proteins. (**C**–**F**) The interaction between recombinant LOXL and recombinant VP19, rVP24, rVP26, or rVP28 was analyzed by GST pull-down assays. The recombinant LOXL could bind to rVP19, rVP24, rVP26, and rVP28. (**G**) The interaction between the Trx-His protein and rVP19, rVP24, rVP26, or rVP28 was analyzed by GST pull-down assays, serving as the negative controls. (**H**–**K**) GST pull-down assays were used to detect the interaction between GST protein and recombinant VP19, rVP24, rVP26, or rVP28. GST protein could not bind to envelope proteins of WSSV.

**Figure 6 viruses-14-02072-f006:**
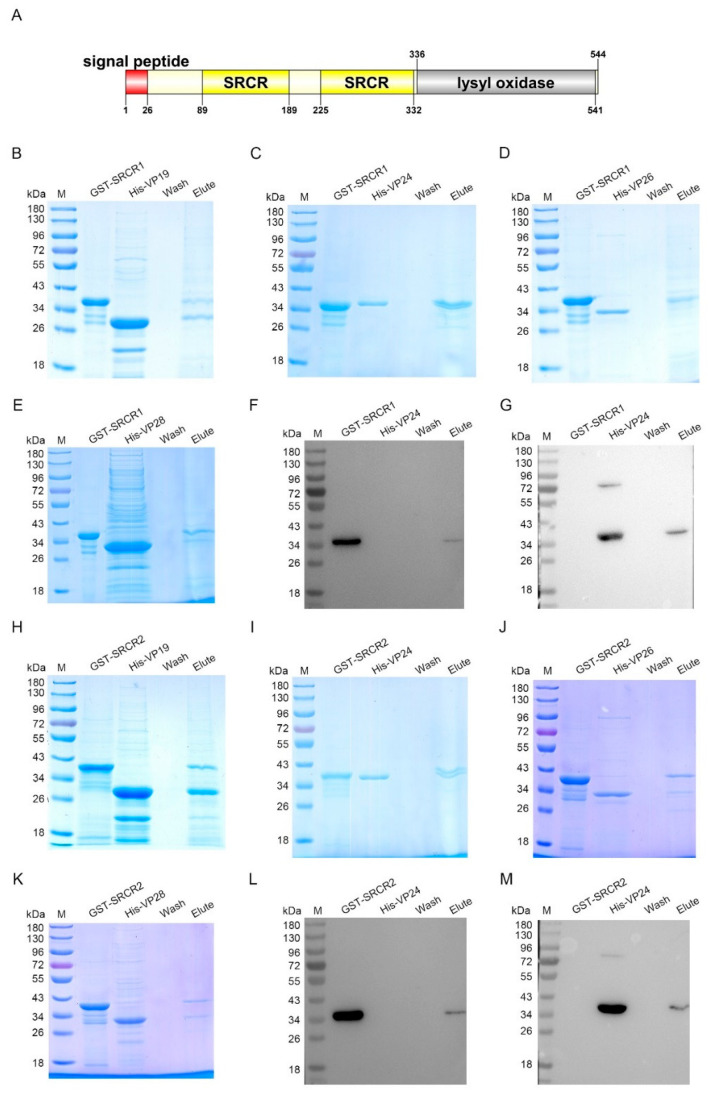
SRCR1 and SRCR2 interacted with envelope proteins of WSSV. (**A**) Domain architecture of LOXL. This result was obtained from the online analysis website, SMART (http://smart.embl-heidelberg.de/ (accessed on 26 October 2020)). (**B**–**E**) The interaction of rSRCR1 with recombinant VP19, rVP24, rVP26, or rVP28 was analyzed by GST pull-down assays. Recombinant SRCR1 could bind to analyzed envelope proteins of WSSV. (**F**,**G**) The interaction between rSRCR1 with rVP24 was analyzed by Western blot using anti-GST antibody (**F**) and anti-HIS antibody (**G**). (**H**–**K**) GST pull-down assays were used to detect the interaction of rSRCR2 with recombinant VP19, rVP24, rVP26, and rVP28. Recombinant SRCR2 could bind to rVP19, rVP24, rVP26, and rVP28. (**L**,**M**) Western blot was used to detect the interaction between rSRCR2 and rVP24 using anti-GST antibody (**L**) and anti-HIS antibody (**M**).

**Figure 7 viruses-14-02072-f007:**
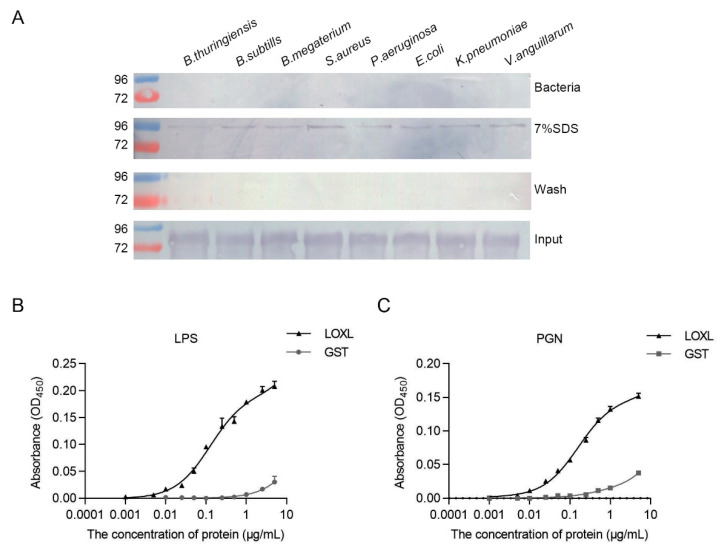
LOXL recognizes bacteria by binding to LPS and PGN. (**A**) Binding activities of rLOXL to different bacteria detected via Western blot using anti-GST antibody as the first antibody. (**B**,**C**) ELISA was performed to detect binding activity of rLOXL to the different polysaccharides, LPS (**B**) and PGN (**C**). All the assays were performed in three independent experiments. The values of the results were expressed as mean ± SD.

**Figure 8 viruses-14-02072-f008:**
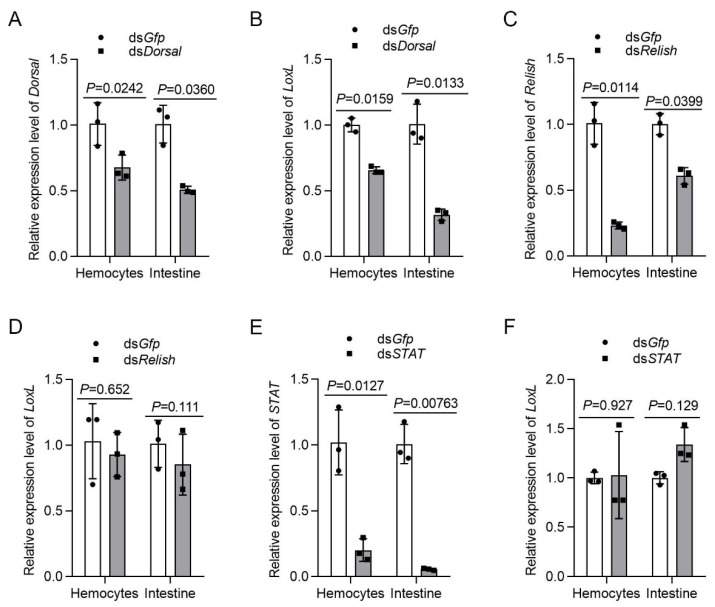
LOXL expression is regulated by Dorsal but not by Relish and Stat. (**A**) The knockdown efficiency of *Dorsal* in hemocytes and intestines by *dsDorsal* injection detected using qPCR; *dsGfp* injection was used as the control. (**B**) Detection of the mRNA level of *LoxL* via qPCR at 24 hpi after *dsDorsal* injection; *dsGfp* injection was used as the control. (**C**) Efficiency of *Relish*-RNAi in the hemocytes and intestines detected via qPCR; *dsGfp* injection was used as the control. (**D**) Expression of mRNA level of *LoxL* detected via qPCR at 24 hpi after *dsRelish* injection; *dsGfp* injection was used as the control. (**E**) Efficiency of *STAT*-RNAi in hemocytes and intestines detected via qPCR; *dsGfp* injection was used as the control. (**F**) Detection of mRNA level of *LoxL* via qPCR at 24 hpi after *dsSTAT* injection; *dsGfp* injection was used as the control. Expression of *β-actin* and *Ef1-α* served as internal reference genes and were processed using geometric means. The values of mRNA relative expression levels were expressed as mean ± SD, and that of the control shrimp was set as 1. Error bars in the figure represent SDs of three replicates.

**Figure 9 viruses-14-02072-f009:**
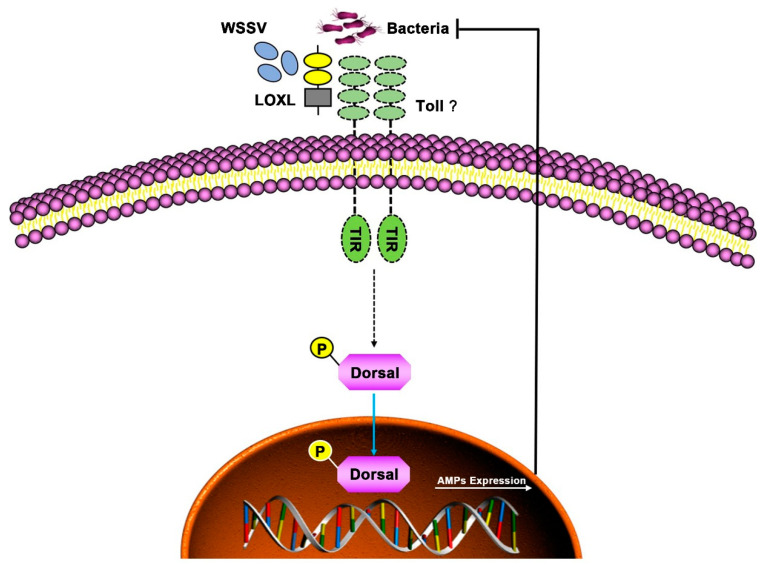
Schematic diagram of LOXL in defending against the infection of WSSV and bacteria as a PRR in shrimp. The LOXL binds to PAMPs of WSSV and bacteria and activates Toll signaling as a co-receptor of Toll receptors, which promotes the Dorsal nuclear translocation and positively regulates the expression of AMPs to resist the infection of pathogens.

## Data Availability

Not applicable.

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
