# Peer review of "Lysyl Oxidase-like Protein Recognizes Viral Envelope Proteins and Bacterial Polysaccharides against Pathogen Infection via Induction of Expression of Antimicrobial Peptides"

_viruses, 2022, doi:10.3390/v14092072_

Round 1

Reviewer 1 Report

The manuscript by Lu and co-authors clearly characterized the antiviral and antibacterial functions of a lysyl oxidase-like (LOXL) protein containing tandem SRCR domains in Marsupenaeus japonicus. The authors provided solid evidence from molecular and organismal levels in support of their findings. Overall the experiment design and related results are presented clearly, and the manuscript was written well.  All findings make great contributions to a better understanding of the innate immunity of invertebrates. 

Author Response

Thanks the positive comments.

Reviewer 2 Report

Review: viruses-1899974

In this work, the authors characterized the antiviral and antibacterial functions of a lysyl oxidase like (LOXL) protein and they have reported that the LOXL plays vital roles in antiviral and antibacterial innate immunity by enhancing the expression of AMPs in shrimp. This manuscript is important and can be published. However, publication of this manuscript in its present form is not recommended. To be considered further for publication, this work will need further clarifications in support of the claims made in the paper. Some specific points of concern are noted below:

1) A schematic illustration of LOXL pathway that describes its mechanistic action and function is necessary.  

2) In this report the authors mentioned that LOXL was involved in the innate immunity of shrimp working as important pattern recognition receptors. The author should be elaborated more on the implications for using LOXL in diagnosis, management or treatment of human diseases.

Minor comments:

1) Suggestion: A model protein structure describing the structure of LOXL will largely improve the quality of the paper.

2) Too many self-citations.

Author Response

Thanks for the positive comments. We also revised our manuscript based on your following comments and suggestions.

1) A schematic illustration of LOXL pathway that describes its mechanistic action and function is necessary. 

Response: Thanks for the suggestion. Following the reviewer’s suggestion, a schematic illustration was added in the revised manuscript, please see Figure 9.

2) In this report the authors mentioned that LOXL was involved in the innate immunity of shrimp working as important pattern recognition receptors. The author should be elaborated more on the implications for using LOXL in diagnosis, management or treatment of human diseases.

Response: Appreciate your comment and suggestion. We have supplemented information about application possibility for LOXL in diagnosis and treatment in human diseases in the discussion part of the revised manuscript (highlighted in yellow).

Minor comments:

1) Suggestion: A model protein structure describing the structure of LOXL will largely improve the quality of the paper.

Response: The panel A in Figure 6 showing domain architecture of LOXL was replaced with a high-resolution picture.

2) Too many self-citations.

Response: We have deleted a reference (Ref. no.29 in original manuscript).